# Self-Supervised Policy Adaptation during Deployment

**Nicklas Hansen**[12]**, Rishabh Jangir**[13]**, Yu Sun**[4]**, Guillem Alenyà**[3]**,
Pieter Abbeel**[4]**, Alexei A Efros**[4]**, Lerrel Pinto**[5]**, Xiaolong Wang**[1]
[1]UC San Diego   [2]Technical University of Denmark
[3]IRI, CSIC-UPC   [4]UC Berkeley   [5]NYU

## Abstract

In most real world scenarios, a policy trained by reinforcement learning in one environment needs to be deployed in another, potentially quite different environment. However, generalization across different environments is known to be hard. A natural solution would be to keep training after deployment in the new environment, but this cannot be done if the new environment offers no reward signal. Our work explores the use of self-supervision to allow the policy to continue training after deployment without using any rewards. While previous methods explicitly anticipate changes in the new environment, we assume no prior knowledge of those changes yet still obtain significant improvements. Empirical evaluations are performed on diverse simulation environments from DeepMind Control suite and ViZDoom, as well as *real* robotic manipulation tasks in continuously changing environments, taking observations from an uncalibrated camera. Our method improves generalization in 31 out of 36 environments across various tasks and outperforms domain randomization on a majority of environments.[1]

## 1 Introduction

Deep reinforcement learning (RL) has achieved considerable success when combined with convolutional neural networks for deriving actions from image pixels (Mnih et al., 2013; Levine et al., 2016; Nair et al., 2018; Yan et al., 2020; Andrychowicz et al., 2020). However, one significant challenge for real-world deployment of vision-based RL remains: a policy trained in one environment might not generalize to other new environments not seen during training. Already hard for RL alone, the challenge is exacerbated when a policy faces high-dimensional visual inputs.

A well explored class of solutions is to learn robust policies that are simply invariant to changes in the environment (Rajeswaran et al., 2016; Tobin et al., 2017; Sadeghi & Levine, 2016; Pinto et al., 2017b; Lee et al., 2019). For example, domain randomization (Tobin et al., 2017; Peng et al., 2018; Pinto et al., 2017a; Yang et al., 2019) applies data augmentation in a simulated environment to train a single robust policy, with the hope that the augmented environment covers enough factors of variation in the test environment. However, this hope may be difficult to realize when the test environment is truly unknown. With too much randomization, training a policy that can simultaneously fit numerous augmented environments requires much larger model and sample complexity. With too little randomization, the actual changes in the test environment might not be covered, and domain randomization may do more harm than good since the randomized factors are now irrelevant. Both phenomena have been observed in our experiments. In all cases, this class of solutions requires human experts to anticipate the changes before the test environment is seen. This cannot scale as more test environments are added with more diverse changes.

Instead of learning a robust policy *invariant* to all possible environmental changes, we argue that it is better for a policy to keep learning during deployment and *adapt* to its actual new environment. A naive way to implement this in RL is to fine-tune the policy in the new environment using rewards as supervision (Rusu et al., 2016; Kalashnikov et al., 2018; Julian et al., 2020). However, while it is relatively easy to craft a dense reward function during training (Gu et al., 2017; Pinto & Gupta, 2016), during deployment it is often impractical and may require substantial engineering efforts.

---

[1]Webpage and implementation: https://nicklashansen.github.io/PAD/

In this paper, we tackle an alternative problem setting in vision-based RL: adapting a pre-trained policy to an unknown environment without any reward. We do this by introducing self-supervision to obtain "free" training signal during deployment. Standard self-supervised learning employs auxiliary tasks designed to automatically create training labels using only the input data (see Section 2 for details). Inspired by this, our policy is jointly trained with two objectives: a standard RL objective and, *additionally*, a self-supervised objective applied on an intermediate representation of the policy network. During training, both objectives are active, maximizing expected reward and simultaneously constraining the intermediate representation through self-supervision. During testing / deployment, only the self-supervised objective (on the raw observational data) remains active, forcing the intermediate representation to adapt to the new environment.

We perform experiments both in simulation and with a real robot. In simulation, we evaluate on two sets of environments: DeepMind Control suite (Tassa et al., 2018) and the CRLMaze ViZDoom (Lomonaco et al., 2019; Wydmuch et al., 2018) navigation task. We evaluate generalization by testing in new environments with visual changes unknown during training. Our method improves generalization in 19 out of 22 test environments across various tasks in DeepMind Control suite, and in all considered test environments on CRLMaze. Besides simulations, we also perform Sim2Real transfer on both reaching and pushing tasks with a Kinova Gen3 robot. After training in simulation, we successfully transfer and adapt policies to 6 different environments, including continuously changing disco lights, on a real robot operating solely from an uncalibrated camera. In both simulation and real experiments, our approach outperforms domain randomization in most environments.

## 2  RELATED WORK

**Self-supervised learning** is a powerful way to learn visual representations from unlabeled data (Vincent et al., 2008; Doersch et al., 2015; Wang & Gupta, 2015; Zhang et al., 2016; Pathak et al., 2016; Noroozi & Favaro, 2016; Zhang et al., 2017; Gidaris et al., 2018). Researchers have proposed to use auxiliary data prediction tasks, such as undoing rotation (Gidaris et al., 2018), solving a jigsaw puzzle (Noroozi & Favaro, 2016), tracking (Wang et al., 2019), etc. to provide supervision in lieu of labels. In RL, the idea of learning visual representations and action at the same time has been investigated (Lange & Riedmiller, 2010; Jaderberg et al., 2016; Pathak et al., 2017; Ha & Schmidhuber, 2018; Yarats et al., 2019; Srinivas et al., 2020; Laskin et al., 2020; Yan et al., 2020). For example, Srinivas et al. (2020) use self-supervised contrastive learning techniques (Chen et al., 2020; Hénaff et al., 2019; Wu et al., 2018; He et al., 2020) to improve sample efficiency in RL by jointly training the self-supervised objective and RL objective. However, this has not been shown to generalize to unseen environments. Other works have applied self-supervision for better generalization across environments (Pathak et al., 2017; Ebert et al., 2018; Sekar et al., 2020). For example, Pathak et al. (2017) use a self-supervised prediction task to provide dense rewards for exploration in novel environments. While results on environment exploration from scratch are encouraging, how to transfer a trained policy (with extrinsic reward) to a novel environment remains unclear. Hence, these methods are not directly applicable to the proposed problem in our paper.

**Generalization across different distributions** is a central challenge in machine learning. In domain adaptation, target domain data is assumed to be accessible (Geirhos et al., 2018; Tzeng et al., 2017; Ganin et al., 2016; Gong et al., 2012; Long et al., 2016; Sun et al., 2019; Julian et al., 2020). For example, Tzeng et al. (2017) use adversarial learning to align the feature representations in both the source and target domain during training. Similarly, the setting of domain generalization (Ghifary et al., 2015; Li et al., 2018; Matsuura & Harada, 2019) assumes that all domains are sampled from the same meta distribution, but the same challenge remains and now becomes generalization across meta-distributions. Our work focuses instead on the setting of generalizing to truly *unseen* changes in the environment which cannot be anticipated at training time.

There have been several recent benchmarks in our setting for image recognition (Hendrycks & Dietterich, 2018; Recht et al., 2018; 2019; Shankar et al., 2019). For example, in Hendrycks & Dietterich (2018), a classifier trained on regular images is tested on corrupted images, with corruption types unknown during training; the method of Hendrycks et al. (2019) is proposed to improve robustness on this benchmark. Following similar spirit, in the context of RL, domain randomization (Tobin et al., 2017; Pinto et al., 2017a; Peng et al., 2018; Ramos et al., 2019; Yang et al., 2019; James et al., 2019) helps a policy trained in simulation to generalize to real robots. For example, Tobin et al. (2017); Sadeghi & Levine (2016) propose to render the simulation environment with random textures and train the policy on top. The learned policy is shown to generalize to real

*Figure 1.* **Left**: Training before deployment. Observations are sampled from a replay buffer for off-policy methods and are collected during roll-outs for on-policy methods. We optimize the RL and self-supervised objectives jointly. **Right**: Policy adaptation during deployment. Observations are collected from the test environment online, and we optimize only the self-supervised objective.

robot manipulation tasks. Instead of deploying a fixed policy, we train and adapt the policy to the new environment with observational data that is naturally revealed during deployment.

**Test-time adaptation for deep learning** is starting to be used in computer vision (Shocher et al., 2017; 2018; Bau et al., 2019; Mullapudi et al., 2019; Sun et al., 2020; Wortsman et al., 2018). For example, Shocher et al. (2018) shows that image super-resolution can be learned at test time (from scratch) simply by trying to upsample a downsampled version of the input image. Bau et al. (2019) show that adapting the prior of a generative adversarial network to the statistics of the test image improves photo manipulation tasks. Our work is closely related to the test-time training method of Sun et al. (2020), which performs joint optimization of image recognition and self-supervised learning with rotation prediction (Gidaris et al., 2018), then uses the self-supervised objective to adapt the representation of individual images during testing. Instead of image recognition, we perform test-time adaptation for RL with visual inputs in an online fashion. As the agent interacts with an environment, we keep obtaining new observational data in a stream for training the visual representations.

## 3 METHOD

In this section, we describe our proposed Policy Adaptation during Deployment (PAD) approach. It can be implemented on top of any policy network and standard RL algorithm (both on-policy and off-policy) that can be described by minimizing some RL objective $J(\theta)$ w.r.t. the collection of parameters $\theta$ using stochastic gradient descent.

### 3.1 NETWORK ARCHITECTURE

We design the network architecture to allow the policy and the self-supervised prediction to share features. For the collection of parameters $\theta$ of a given policy network $\pi$, we split it sequentially into $\theta = (\theta_e, \theta_a)$, where $\theta_e$ collects the parameters of the feature extractor, and $\theta_a$ is the head that outputs a distribution over actions. We define networks $\pi_e$ with parameters $\theta_e$ and $\pi_a$ with parameters $\theta_a$ such that $\pi(\mathbf{s}; \theta) = \pi_a(\pi_e(\mathbf{s}))$, where $\mathbf{s}$ represents an image observation. Intuitively, one can think of $\pi_e$ as a feature extractor, and $\pi_a$ as a controller based on these features. The goal of our method is to update $\pi_e$ at test-time using gradients from a self-supervised task, such that $\pi_e$ (and consequently $\pi_\theta$) can generalize. Let $\pi_s$ with parameters $\theta_s$ be the self-supervised prediction head and its collection of parameters, and the input to $\pi_s$ be the output of $\pi_e$ (as illustrated in Figure 1). In this work, the self-supervised task is inverse dynamics prediction for control, and rotation prediction for navigation.

### 3.2 INVERSE DYNAMICS PREDICTION AND ROTATION PREDICTION

At each time step, we always observe a transition sequence in the form of $(\mathbf{s}_t, \mathbf{a}_t, \mathbf{s}_{t+1})$, during both training and testing. Naturally, self-supervision can be derived from taking parts of the sequence and predicting the rest. An inverse dynamics model takes the states before and after transition, and predicts the action in between. In this work, the inverse dynamics model $\pi_s$ operates on the feature space extracted by $\pi_e$. We can write the inverse dynamics prediction objective formally as

$$L(\theta_s, \theta_e) = \ell\big(\mathbf{a}_t, \ \pi_s\left(\pi_e(\mathbf{s}_t), \pi_e(\mathbf{s}_{t+1})\right)\big). \tag{1}$$

For continuous actions, $\ell$ is the mean squared error between the ground truth and the model output. For discrete actions, the output is a soft-max distribution over the action space, and $\ell$ is the cross-

entropy loss. Empirically, we find this self-supervised task to be most effective with continuous actions, possibly because inverse dynamics prediction in a small space of discrete actions is not as challenging. Note that we predict the inverse dynamics instead of the forward dynamics, because when operating in feature space, the latter can produce trivial solutions such as the constant zero feature for every state[2]. If we instead performed prediction with forward dynamics in pixel space, the task would be extremely challenging given the large uncertainty in pixel prediction.

As an alternative self-supervised task, we use rotation prediction (Gidaris et al., 2018). We rotate an image by one of 0, 90, 180 and 270 degrees as input to the network, and cast this as a four-way classification problem to determine which one of these four ways the image has been rotated. This task is shown to be effective for learning representations for object configuration and scene structure, which is beneficial for visual recognition (Hendrycks et al., 2019; Doersch & Zisserman, 2017).

### 3.3 TRAINING AND TESTING

Before deployment of the policy, because we have signals from both the reward and self-supervised auxiliary task, we can train with both in the fashion of multi-task learning. This corresponds to the following optimization problem during training $\min_{\theta_a, \theta_s, \theta_e} J(\theta_a, \theta_e) + \alpha L(\theta_s, \theta_e)$, where $\alpha > 0$ is a trade-off hyperparameter. During deployment, we cannot optimize $J$ anymore since the reward is unavailable, but we can still optimize $L$ to update both $\theta_s$ and $\theta_e$. Empirically, we find only negligible difference with keeping $\theta_s$ fixed at test-time, so we update both since the gradients have to be computed regardless; we ablate this decision in appendix C. As we obtain new images from the stream of visual inputs in the environment, $\theta$ keeps being updated until the episode ends. This corresponds to, for each iteration $t = 1...T$:

$$\mathbf{s}_t \sim p(\mathbf{s}_t | \mathbf{a}_{t-1}, \mathbf{s}_{t-1}) \tag{2}$$
$$\theta_s(t) = \theta_s(t-1) - \nabla_{\theta_s} L(\mathbf{s}_t; \theta_s(t-1), \theta_e(t-1)) \tag{3}$$
$$\theta_e(t) = \theta_e(t-1) - \nabla_{\theta_e} L(\mathbf{s}_t; \theta_s(t-1), \theta_e(t-1)) \tag{4}$$
$$\mathbf{a}_t = \pi(\mathbf{s}_t; \theta(t)) \quad \text{with} \quad \theta(t) = (\theta_e(t), \theta_a), \tag{5}$$

where $\theta_s(0) = \theta_s, \theta_e(0) = \theta_e, \mathbf{s}_0$ is the initial condition given by the environment, $\mathbf{a}_0 = \pi_\theta(\mathbf{s}_0)$, $p$ is the unknown environment transition, and $L$ is the self-supervised objective as previously introduced.

## 4 EXPERIMENTS

In this work, we investigate how well an agent trained in one environment (denoted the *training environment*) generalizes to *unseen* and diverse test environments. During evaluation, agents have no access to reward signals and are expected to generalize without trials nor prior knowledge about the test environments. In simulation, we evaluate our method (PAD) and baselines extensively on continuous control tasks from DeepMind Control (DMControl) suite (Tassa et al., 2018) as well as the CRLMaze (Lomonaco et al., 2019) navigation task, and experiment with both stationary (colors, objects, textures, lighting) and non-stationary (videos) environment changes. We further show that PAD transfers from simulation to a real robot and successfully adapts to environmental differences during deployment in two robotic manipulation tasks. Samples from DMControl and CRLMaze environments are shown in Figure 2, and samples from the robot experiments are shown in Figure 4. Implementation is available at `https://nicklashansen.github.io/PAD/`.

**Network details.** For DMControl and the robotic manipulation tasks we implement PAD on top of Soft Actor-Critic (SAC) (Haarnoja et al., 2018), and adopt both network architecture and hyperparameters from Yarats et al. (2019), with minor modifications: the feature extractor $\pi_e$ has 8 convolutional layers shared between the RL head $\pi_a$ and self-supervised head $\pi_s$, and we split the network into architecturally identical heads following $\pi_e$. Each head consists of 3 convolutional layers followed by 4 fully connected layers. For CRLMaze, we use Advantage Actor-Critic (A2C) as base algorithm (Mnih et al., 2016) and apply the same architecture as for the other experiments, but implement $\pi_e$ with only 6 convolutional layers. Observations are stacks of $k$ colored frames ($k = 3$ on DMControl and CRLMaze; $k = 1$ in robotic manipulation) of size $100 \times 100$ and time-consistent random crop is applied as in Srinivas et al. (2020). During deployment, we optimize the self-supervised objective online w.r.t. $\theta_e, \theta_s$ for one gradient step per time iteration. See appendix F for implementation details.

---

[2]A forward dynamics model operating in feature space can trivially achieve a loss of 0 by learning to map every state to a constant vector, e.g. $\mathbf{0}$. An inverse dynamics model, however, does not have such trivial solutions.

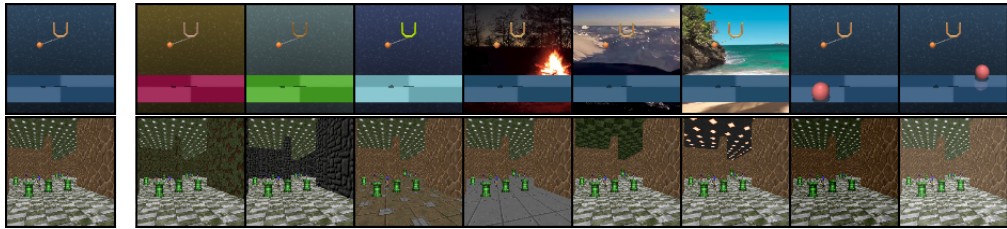

*Figure 2.* **Left:** Training environments of DMControl (top) and CRLMaze (bottom). **Right:** Test environments of DMControl (top) and CRLMaze (bottom). Changes to DMControl include randomized colors, video backgrounds, and distractors; changes to CRLMaze include textures and lighting.

*Table 1.* Episodic return in test environments with randomized colors, mean and std. dev. for 10 seeds. Best method on each task is in bold and blue compares SAC+IDM with and without PAD.

| | | | | | 10x episode length | |
| Random colors | SAC | +DR | +IDM | +IDM (PAD) | +IDM | +IDM (PAD) |
|---|---|---|---|---|---|---|
| Walker, walk | $414_{\pm 74}$ | $\mathbf{594_{\pm 104}}$ | $406_{\pm 29}$ | $468_{\pm 47}$ | $3830_{\pm 547}$ | $\mathbf{5505_{\pm 592}}$ |
| Walker, stand | $719_{\pm 74}$ | $715_{\pm 96}$ | $743_{\pm 37}$ | $\mathbf{797_{\pm 46}}$ | $7832_{\pm 209}$ | $\mathbf{8566_{\pm 121}}$ |
| Cartpole, swingup | $592_{\pm 50}$ | $\mathbf{647_{\pm 48}}$ | $585_{\pm 73}$ | $630_{\pm 63}$ | $6528_{\pm 539}$ | $\mathbf{7093_{\pm 592}}$ |
| Cartpole, balance | $857_{\pm 60}$ | $\mathbf{867_{\pm 37}}$ | $835_{\pm 40}$ | $848_{\pm 29}$ | $\mathbf{7746_{\pm 526}}$ | $7670_{\pm 293}$ |
| Ball in cup, catch | $411_{\pm 183}$ | $470_{\pm 252}$ | $471_{\pm 75}$ | $\mathbf{563_{\pm 50}}$ | – | – |
| Finger, spin | $626_{\pm 163}$ | $465_{\pm 314}$ | $757_{\pm 62}$ | $\mathbf{803_{\pm 72}}$ | $7249_{\pm 642}$ | $\mathbf{7496_{\pm 655}}$ |
| Finger, turn_easy | $270_{\pm 43}$ | $167_{\pm 26}$ | $283_{\pm 51}$ | $\mathbf{304_{\pm 46}}$ | – | – |
| Cheetah, run | $154_{\pm 41}$ | $145_{\pm 29}$ | $121_{\pm 38}$ | $\mathbf{159_{\pm 28}}$ | $1117_{\pm 530}$ | $\mathbf{1208_{\pm 487}}$ |
| Reacher, easy | $163_{\pm 45}$ | $105_{\pm 37}$ | $201_{\pm 32}$ | $\mathbf{214_{\pm 44}}$ | $1788_{\pm 441}$ | $\mathbf{2152_{\pm 506}}$ |

## 4.1 DEEPMIND CONTROL

DeepMind Control (DMControl) (Tassa et al., 2018) is a collection of continuous control tasks where agents only observe raw pixels. Generalization benchmarks on DMControl represent diverse real-world tasks for motor control, and contain distracting surroundings not correlated with the reward signals.

**Experimental setup.** We experiment with 9 tasks from DM-Control and measure generalization to four types of test environments: (i) randomized colors; (ii) natural videos as background; (iii) distracting objects placed in the scene; and (iv) the unmodified training environment. For each test environment, we evaluate methods across 10 seeds and 100 random initializations. If a given test environment is not applicable to certain tasks, e.g. if a task has no background for the video background setting, they are excluded. Tasks are selected on the basis of diversity, as well as the success of vision-based RL in prior work (Yarats et al., 2019; Srinivas et al., 2020; Laskin et al., 2020; Kostrikov et al., 2020). We implement PAD on top of SAC and use an Inverse Dynamics Model (IDM) for self-supervision, as we find that learning a model of the dynamics works well for motor control. For completeness, we ablate the choice of self-supervision. Learning curves are provided in appendix B.

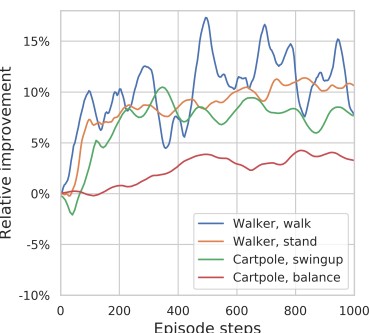

*Figure 3.* Relative improvement in instantaneous reward over time for PAD on the random color env.

We compare our method to the following baselines: (i) SAC with no changes (denoted *SAC*); (ii) SAC trained with domain randomization on a fixed set of 100 colors (denoted *+DR*); and (iii) SAC trained jointly with an IDM but without PAD (denoted *+IDM*). Our method using an IDM with PAD is denoted by *+IDM (PAD)*. For domain randomization, colors are sampled from the *same distribution* as in evaluation, but with lower variance, as we find that training directly on the test distribution does not converge.

**Random perturbation of color.** Robustness to subtle changes such as color is essential to real-world deployment of RL policies. We evaluate generalization on a fixed set of 100 colors of foreground, background and the agent itself, and report the results in Table 1 (first 4 columns). We find PAD to improve generalization *in all tasks considered*, outperforming SAC trained with domain

randomization in **6** out of **9** tasks. Surprisingly, despite a substantial overlap between training and test domains of domain randomization, it generalizes no better than vanilla SAC on a majority of tasks.

**Long-term stability.** We find the relative improvement of PAD to improve over time, as shown in Figure 3. To examine the long-term stability of PAD, we further evaluate on 10x episode lengths and summarize the results in the last two columns in Table 1 (goal-oriented tasks excluded). While we do not explicitly prevent the embedding from drifting away from the RL task, we find empirically that PAD does not degrade the performance of the policy, even over long horizons, and when PAD does *not* improve, we find it to hurt minimally. We conjecture this is because we are not learning a new task, but simply continue to optimize the same (self-supervised) objective as during joint training, where both two tasks are compatible. In this setting, PAD still improves generalization in **6** out of **7** tasks, and thus naturally extends beyond episodic deployment. For completeness, we also evaluate methods in the environment in which they were trained, and report the results in appendix A. We find that, while PAD improves generalization to novel environments, performance is virtually unchanged on the training environment. We conjecture this is because the self-supervised task is already fully learned and any continued training on the same data distribution thus has little impact.

**Non-stationary environments.** To investigate whether PAD can adapt in non-stationary environments, we evaluate generalization to diverse video backgrounds (refer to Figure 2). We find PAD to outperform all baselines on **7** out of **8** tasks, as shown in Table 2, by as much as **104%** over domain randomization on *Finger, spin*. Domain randomization generalizes comparably worse to videos, which we conjecture is not because the environments are non-stationary, but rather because the image statistics of videos are not covered by its training domain of randomized colors. In fact, domain randomization is outperformed by the vanilla SAC in most tasks with video backgrounds, which is in line with the findings of Packer et al. (2018).

*Table 2.* Episodic return in test environments with video backgrounds (top) and distracting objects (bottom), mean and std. dev. for 10 seeds. Best method on each task is in bold and blue compares SAC+IDM with and without PAD.

| Video backgrounds | SAC | +DR | +IDM | +IDM (PAD) |
|---|---|---|---|---|
| Walker, walk | $616\pm80$ | $655\pm55$ | $694\pm85$ | $\mathbf{717\pm79}$ |
| Walker, stand | $899\pm53$ | $869\pm60$ | $902\pm51$ | $\mathbf{935\pm20}$ |
| Cartpole, swingup | $375\pm90$ | $485\pm67$ | $487\pm90$ | $\mathbf{521\pm76}$ |
| Cartpole, balance | $693\pm109$ | $\mathbf{766\pm92}$ | $691\pm76$ | $687\pm58$ |
| Ball in cup, catch | $393\pm175$ | $271\pm189$ | $362\pm69$ | $\mathbf{436\pm55}$ |
| Finger, spin | $447\pm102$ | $338\pm207$ | $605\pm61$ | $\mathbf{691\pm80}$ |
| Finger, turn_easy | $355\pm108$ | $223\pm91$ | $355\pm110$ | $\mathbf{362\pm101}$ |
| Cheetah, run | $194\pm30$ | $150\pm34$ | $164\pm42$ | $\mathbf{206\pm34}$ |

| Distracting objects | SAC | +DR | +IDM | +IDM (PAD) |
|---|---|---|---|---|
| Cartpole, swingup | $\mathbf{815\pm60}$ | $809\pm24$ | $776\pm58$ | $771\pm64$ |
| Cartpole, balance | $\mathbf{969\pm20}$ | $938\pm35$ | $964\pm26$ | $960\pm29$ |
| Ball in cup, catch | $177\pm111$ | $331\pm189$ | $482\pm128$ | $\mathbf{545\pm173}$ |
| Finger, spin | $652\pm184$ | $564\pm288$ | $836\pm62$ | $\mathbf{867\pm72}$ |
| Finger, turn_easy | $302\pm68$ | $165\pm12$ | $326\pm101$ | $\mathbf{347\pm48}$ |

**Scene content.** We hypothesize that: (i) an agent trained with an IDM is comparably less distracted by scene content since objects uncorrelated to actions yield no predictive power; and (ii) that PAD can adapt to unexpected objects in the scene. We test these hypotheses by measuring robustness to colored shapes at a variety of positions in both the foreground and background of the scene (no physical interaction). Results are summarized in Table 2. PAD outperforms all baselines in **3** out of **5** tasks, with a relative improvement of **208%** over SAC on *Ball in cup, catch*. In the two cartpole tasks in which PAD does not improve, all methods are already relatively unaffected by the distractors.

**Choice of self-supervised task.** We investigate how much the choice of self-supervised task contributes to the overall success of our method, and consider the following ablations: (i) replacing inverse dynamics with the rotation prediction task described in Section 3.2; and (ii) replacing it with the recently proposed CURL (Srinivas et al., 2020) contrastive learning algorithm for RL. As shown in Table 3, PAD improves generalization of CURL in a majority of tasks on the randomized color benchmark, and in 4 out of 9 tasks using rotation prediction. However, inverse dynamics as auxiliary task produces more consistent results and offers better generalization overall. We argue that learning an IDM produces better representations for motor control since it connects observations directly to actions, whereas CURL and rotation prediction operates purely on observations. In general, we find the improvement of PAD to be bigger in tasks that benefit significantly from visual information (see appendix A), and conjecture that selecting a self-supervised task that learns features useful to the RL task is crucial to the success of PAD, which we discuss further in Section 4.2.

*Table 3.* Ablations on the randomized color domain of DMC. All methods use SAC. CURL represents RL with a contrastive learning task (Srinivas et al., 2020) and Rot represents the rotation prediction (Gidaris et al., 2018). Offline PAD is here denoted O-PAD for brevity, whereas the default usage of PAD is in an online setting. Best method is in bold and blue compares +IDM w/ and w/o PAD.

| Random colors | CURL | CURL (PAD) | Rot | Rot (PAD) | IDM | IDM (O-PAD) | IDM (PAD) |
|---|---|---|---|---|---|---|---|
| Walker, walk | $445_{\pm99}$ | $\mathbf{495_{\pm70}}$ | $335_{\pm7}$ | $330_{\pm30}$ | $406_{\pm29}$ | $441_{\pm16}$ | $468_{\pm47}$ |
| Walker, stand | $662_{\pm54}$ | $753_{\pm49}$ | $673_{\pm4}$ | $653_{\pm27}$ | $743_{\pm37}$ | $727_{\pm21}$ | $\mathbf{797_{\pm46}}$ |
| Cartpole, swingup | $454_{\pm110}$ | $413_{\pm67}$ | $493_{\pm52}$ | $477_{\pm38}$ | $585_{\pm73}$ | $578_{\pm69}$ | $\mathbf{630_{\pm63}}$ |
| Cartpole, balance | $782_{\pm13}$ | $763_{\pm5}$ | $710_{\pm72}$ | $734_{\pm81}$ | $835_{\pm40}$ | $796_{\pm37}$ | $\mathbf{848_{\pm29}}$ |
| Ball in cup, catch | $231_{\pm92}$ | $332_{\pm78}$ | $291_{\pm54}$ | $314_{\pm60}$ | $471_{\pm75}$ | $490_{\pm16}$ | $\mathbf{563_{\pm50}}$ |
| Finger, spin | $691_{\pm12}$ | $588_{\pm22}$ | $695_{\pm36}$ | $689_{\pm20}$ | $757_{\pm62}$ | $767_{\pm43}$ | $\mathbf{803_{\pm72}}$ |
| Finger, turn_easy | $202_{\pm32}$ | $186_{\pm2}$ | $283_{\pm68}$ | $230_{\pm53}$ | $283_{\pm51}$ | $\mathbf{321_{\pm10}}$ | $304_{\pm46}$ |
| Cheetah, run | $202_{\pm22}$ | $\mathbf{211_{\pm20}}$ | $127_{\pm3}$ | $135_{\pm12}$ | $121_{\pm38}$ | $112_{\pm35}$ | $159_{\pm28}$ |
| Reacher, easy | $325_{\pm32}$ | $\mathbf{378_{\pm62}}$ | $99_{\pm29}$ | $120_{\pm7}$ | $201_{\pm32}$ | $241_{\pm24}$ | $214_{\pm44}$ |

*Table 4.* Episodic return of PAD and baselines in CRLMaze environments. PAD improves generalization *in all considered environments* and outperforms both A2C and domain randomization by a large margin. All methods use A2C. We report mean and std. error of 10 seeds. Best method in each environment is in bold and blue compares rotation prediction with and without PAD.

| CRLMaze | Random | A2C | +DR | +IDM | +IDM (PAD) | +Rot | +Rot (PAD) |
|---|---|---|---|---|---|---|---|
| Walls | $-870_{\pm30}$ | $-380_{\pm145}$ | $-260_{\pm137}$ | $-302_{\pm150}$ | $-428_{\pm135}$ | $-206_{\pm166}$ | $\mathbf{-74_{\pm116}}$ |
| Floor | $-868_{\pm23}$ | $-320_{\pm167}$ | $-438_{\pm59}$ | $\mathbf{-47_{\pm198}}$ | $-530_{\pm106}$ | $-294_{\pm123}$ | $-209_{\pm94}$ |
| Ceiling | $-872_{\pm30}$ | $-171_{\pm175}$ | $-400_{\pm74}$ | $166_{\pm215}$ | $-508_{\pm104}$ | $128_{\pm196}$ | $\mathbf{281_{\pm83}}$ |
| Lights | $-900_{\pm29}$ | $-30_{\pm213}$ | $-310_{\pm106}$ | $239_{\pm270}$ | $-460_{\pm114}$ | $-84_{\pm53}$ | $\mathbf{312_{\pm104}}$ |

**Offline versus online learning.** Observations that arrive sequentially are highly correlated, and we thus hypothesize that our method benefits significantly from learning online. To test this hypothesis, we run an *offline* variant of our method in which network updates are forgotten after each step. In this setting, our method can only adapt to single observations and does not benefit from learning over time. Results are shown in Table 3. We find that our method benefits substantially from online learning, but learning offline still improves generalization on select tasks.

## 4.2 CRLMAZE

CRLMaze (Lomonaco et al., 2019) is a time-constrained, discrete-action 3D navigation task for ViZDoom (Wydmuch et al., 2018), in which an agent is to navigate a maze and collect objects. There is a positive reward associated with green columns, and a negative reward for lanterns as well as for living. Readers are referred to the respective papers for details on the task and environment.

**Experimental setup.** We train agents on a single environment and measure generalization to environments with novel textures for walls, floor, and ceiling, as well as lighting, as shown in Figure 2. We implement PAD on top of A2C (Mnih et al., 2016) and use rotation prediction (see Section 3.2) as self-supervised task. Learning to navigate novel scenes requires a generalized scene understanding, and we find that rotation prediction facilitates that more so than an IDM. We compare to the following baselines: (i) a random agent (denoted *Random*); (ii) A2C with no changes (denoted *A2C*); (iii) A2C trained with domain randomization (denoted *+DR*); (iv) A2C with an IDM as auxiliary task (denoted *+IDM*); and (v) A2C with rotation prediction as auxiliary task (denoted *+Rot*). We denote Rot with PAD as *+Rot (PAD)*. Domain randomization uses 56 combinations of diverse textures, partially overlapping with the test distribution, and we find it necessary to train domain randomization for twice as many episodes in order to converge. We closely follow the evaluation procedure of (Lomonaco et al., 2019) and evaluate methods across 20 starting positions and 10 random seeds.

**Results.** We report performance on the CRLMaze environments in Table 4. PAD improves generalization *in all considered test environments*, outperforming both A2C and domain randomization by a large margin. Domain randomization performs consistently across all environments but is less successful overall. We further examine the importance of selecting appropriate auxiliary tasks by a simple ablation: replacing rotation prediction with an IDM for the navigation task. We conjecture that, while an auxiliary task can enforce structure in the learned representations, its features (and consequently gradients) need to be sufficiently correlated with the primary RL task for PAD to be

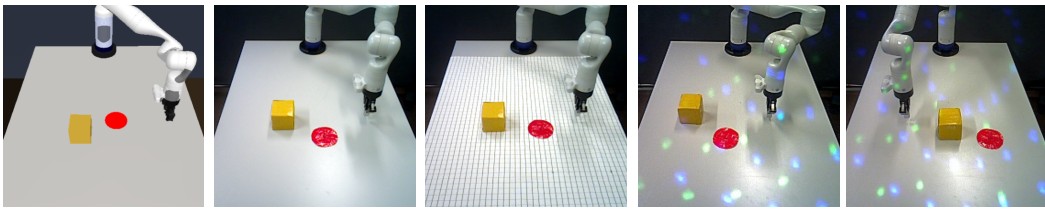

*(a)* Simulation.  *(b)* Default transfer.  *(c)* Table cloth.  *(d)* Disco lights.

*Figure 4.* Samples from the *push* robotic manipulation task. The task is to push the yellow cube to the location of the red disc. Agents are trained in setting (a) and evaluated in settings (b-d).

successful during deployment. While PAD with rotation prediction improves generalization across all test environments considered, IDM does not, which suggests that rotation prediction is more suitable for tasks that require scene understanding, whereas IDM is useful for tasks that require motor control. We leave it to future work to automate the process of selecting appropriate auxiliary tasks.

## 4.3 ROBOTIC MANIPULATION TASKS

We deploy our method and baselines on a real Kinova Gen3 robot and evaluate on two manipulation tasks: (i) *reach*, a task in which the robot reaches for a goal marked by a red disc; and (ii) *push*, a task in which the robot pushes a cube to the location of the red disc. Both tasks use an XY action space, where the Z position of the actuator is fixed. Agents operate purely from pixel observations with *no access to state information*. During deployment, we make no effort to calibrate camera, lighting, or physical properties such as dimensions, mass, and friction, and policies are expected to generalize with no prior knowledge of the test environment. Samples from the *push* task are shown in Figure 4, and samples from *reach* are shown in appendix E.

*Table 5.* Success rate of PAD and baselines on a *real* robotic arm. Best method in each environment is in bold and blue compares +IDM with and without PAD.

| Real robot | SAC | +DR | +IDM | +IDM (PAD) |
|---|---|---|---|---|
| Reach (default) | 100% | 100% | 100% | 100% |
| Reach (cloth) | 48% | **80%** | 56% | **80%** |
| Reach (disco) | 72% | 76% | 88% | **92%** |
| Push (default) | 88% | 88% | 92% | **100%** |
| Push (cloth) | 60% | 64% | 64% | **88%** |
| Push (disco) | 60% | 68% | 72% | **84%** |

**Experimental setup.** We implement PAD on top of SAC (Haarnoja et al., 2018) and apply the same experimental setup as in Section 4.1 using an Inverse Dynamics Model (IDM) for self-supervision, but without frame-stacking (i.e. $k = 1$). Agents are trained in simulation with dense rewards and randomized initial configurations of arm, goal, and box, and we measure generalization to 3 novel environments in the real-world: (i) default environment with pixel observations that roughly mimic the simulation; (ii) a patterned table cloth that

*Table 6.* Success rate of PAD and baselines for the *push* task on a *simulated* robotic arm in test environments with changes to *dynamics*. Changes include object mass, size, and friction, arm mount position, and end effector velocity. Best method in each environment is in bold and blue compares +IDM with and without PAD.

| Simulated robot | SAC | +DR | +IDM | +IDM (PAD) |
|---|---|---|---|---|
| Push (object) | 66% | 64% | 72% | **82%** |
| Push (mount) | 68% | 58% | **86%** | 84% |
| Push (velocity) | 70% | 68% | 70% | **78%** |
| Push (all) | 56% | 50% | 48% | **76%** |

distracts visually and greatly increases friction; and (iii) disco, an environment with non-stationary visual disco light distractions. Notably, all 3 environments also feature subtle differences in dynamics compared to the training environment, such as object dimensions, mass, friction, and uncalibrated actions. In each setting, we evaluate the success rate across 25 test runs spanning across 5 pre-defined goal locations throughout the table. The goal locations vary between the two tasks, and the robot is reset after each run. We perform comparison against direct transfer and domain randomization baselines as in Section 4.1. We further evaluate generalization to changes in dynamics by considering a variant of the simulated environment in which object mass, size, and friction, arm mount position, and end effector velocity is modified. We consider each setting both individually and jointly, and evaluate success rate across 50 unique configurations with the robot reset after each run.

**Results.** We report transfer results in Table 5. While all methods transfer successfully to *reach (default)*, we observe PAD to improve generalization in all settings in which the baselines show

sub-optimal performance. We find PAD to be especially powerful for the *push* task that involves dynamics, improving by as much as **24%** in *push (cloth)*. While domain randomization proves highly effective in *reach (cloth)*, we observe no significant benefit in the other settings, which suggests that PAD can be more suitable in challenging tasks like *push*. To isolate the effect of dynamics, we further evaluate generalization to a number of simulated changes in dynamics on the *push* task. Results are shown in Table 6. We find PAD to improve generalization to changes in the physical properties of the object and end effector, whereas both *SAC+IDM* and PAD are relatively unaffected by changes to the mount position. Consistent with the real robot results in Section 5, PAD is found to be most effective when changes in dynamics are non-trivial, improving by as much as **28%** in the *push (all)* setting, where all 3 environmental changes are considered jointly. These results suggest that PAD can be a simple, yet effective method for generalization to diverse, unseen environments that vary in both visuals and dynamics.

## 5 CONCLUSION

While previous work addresses generalization in RL by learning policies that are invariant to any environment changes that can be anticipated, we formulate an alternative problem setting in vision-based RL: can we instead *adapt* a pretrained-policy to new environments without any reward. We propose Policy Adaptation during Deployment, a self-supervised framework for online adaptation at test-time, and show empirically that our method improves generalization of policies to diverse simulated and real-world environmental changes across a variety of tasks. We find our approach benefits greatly from learning online, and we systematically evaluate how the choice of self-supervised task impacts performance. While the current framework relies on prior knowledge on selecting self-supervised tasks for policy adaptation, we see our work as the initial step in addressing the problem of adapting vision-based policies to unknown environments. We ultimately envision embodied agents in the future to be learning all the time, with the flexibility to learn both with and without rewards, before and during deployment.

**Acknowledgements.** This work was supported, in part, by grants from DARPA, NSF 1730158 CI-New: Cognitive Hardware and Software Ecosystem Community Infrastructure (CHASE-CI), NSF ACI-1541349 CC*DNI Pacific Research Platform, and gifts from Qualcomm and TuSimple. This work was also funded, in part, by grants from Berkeley DeepDrive, SAP and European Research Council (ERC) from the European Union Horizon 2020 Programme under grant agreement no. 741930 (CLOTHILDE). We would like to thank Fenglu Hong and Joey Hejna for helpful discussions.

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

## A    PERFORMANCE ON THE TRAINING ENVIRONMENT

Historically, agents have commonly been trained and evaluated in the same environment when benchmarking RL algorithms exclusively in simulation. Although such an evaluation procedure does not consider generalization, it is still a useful metric for comparison of sample efficiency and stability of algorithms. For completeness, we also evaluate our method and baselines in this setting on both DMControl and CRLMaze. DMControl results are reported in Table 7 and results on the CRLMaze environment are shown in Table 8. In this setting, we also compare to an additional baseline on DMControl: a blind SAC agent that operates purely on its previous actions. The performance of a blind agent indicates to which degree a given task benefits from visual information. We find that, while PAD improves generalization to novel environments, performance is virtually unchanged when evaluated on the same environment as in training. We conjecture that this is because the algorithm already is adapted to the training environment and any continued training on the same data distribution thus has little influence. We further emphasize that, even when evaluated on the training environment, PAD still outperforms baselines on most tasks. For example, we observe a **15%** relative improvement over SAC on the *Finger, spin* task. We hypothesize that this gain in performance is because the self-supervised objective improves learning by constraining the intermediate representation of policies. A blind agent is no better than random on this particular task, which would suggest that agents benefit substantially from visual information in *Finger, spin*. Therefore, learning a good intermediate representation of that information is highly beneficial to the RL objective, which we find PAD to facilitate through its self-supervised learning framework. Likewise, the SAC baseline only achieves a 51% improvement over the blind agent on *Cartpole, balance*, which indicates that extracting visual information from observations is not as crucial on this task. Consequently, both PAD and baselines achieve similar performance on this task.

*Table 7.* Episodic return on the training environment for each of the 9 tasks considered in DMControl, mean and std. dev. for 10 seeds. Best method on each task is in bold and blue compares +IDM with and without PAD. It is shown that PAD hurts minimally when the environment is unchanged.

| Training env. | Blind | SAC | +DR | +IDM | +IDM (PAD) |
|---|---|---|---|---|---|
| Walker, walk | $235_{\pm17}$ | $847_{\pm71}$ | $756_{\pm71}$ | $\mathbf{911_{\pm24}}$ | $895_{\pm28}$ |
| Walker, stand | $388_{\pm10}$ | $959_{\pm11}$ | $928_{\pm36}$ | $\mathbf{966_{\pm8}}$ | $956_{\pm20}$ |
| Cartpole, swingup | $132_{\pm41}$ | $\mathbf{850_{\pm28}}$ | $807_{\pm36}$ | $849_{\pm30}$ | $845_{\pm34}$ |
| Cartpole, balance | $646_{\pm131}$ | $978_{\pm22}$ | $971_{\pm30}$ | $\mathbf{982_{\pm20}}$ | $979_{\pm21}$ |
| Ball in cup, catch | $150_{\pm96}$ | $725_{\pm355}$ | $469_{\pm339}$ | $\mathbf{919_{\pm118}}$ | $910_{\pm129}$ |
| Finger, spin | $3_{\pm2}$ | $809_{\pm138}$ | $686_{\pm295}$ | $\mathbf{928_{\pm45}}$ | $927_{\pm45}$ |
| Finger, turn_easy | $172_{\pm27}$ | $\mathbf{462_{\pm146}}$ | $243_{\pm124}$ | $\mathbf{462_{\pm152}}$ | $455_{\pm160}$ |
| Cheetah, run | $264_{\pm75}$ | $\mathbf{387_{\pm74}}$ | $195_{\pm46}$ | $384_{\pm88}$ | $380_{\pm91}$ |
| Reacher, easy | $107_{\pm11}$ | $264_{\pm113}$ | $92_{\pm45}$ | $\mathbf{390_{\pm126}}$ | $365_{\pm114}$ |

*Table 8.* Episodic return of PAD and baselines in the CRLMaze training environment. All methods use A2C. We report mean and std. error of 10 seeds. Best method is in bold and blue compares rotation prediction with and without PAD.

| CRLMaze | Random | A2C | +DR | +IDM | +IDM (PAD) | +Rot | +Rot (PAD) |
|---|---|---|---|---|---|---|---|
| Training env. | $-868_{\pm34}$ | $371_{\pm198}$ | $-355_{\pm93}$ | $585_{\pm246}$ | $-416_{\pm135}$ | $\mathbf{729_{\pm148}}$ | $681_{\pm99}$ |

## B    LEARNING CURVES ON DEEPMIND CONTROL

All methods are trained until convergence (500,000 frames) on DMControl. While we do not consider the sample efficiency of our method and baselines in this study, we report learning curves for SAC, SAC+IDM and SAC trained with domain randomization on three tasks in Figure 5 for completeness. SAC trained with and without an IDM are similar in terms of sample efficiency and final performance, whereas domain randomization consistently displays worse sample efficiency, larger variation between seeds, and converges to sub-optimal performance in two out of the three tasks shown.

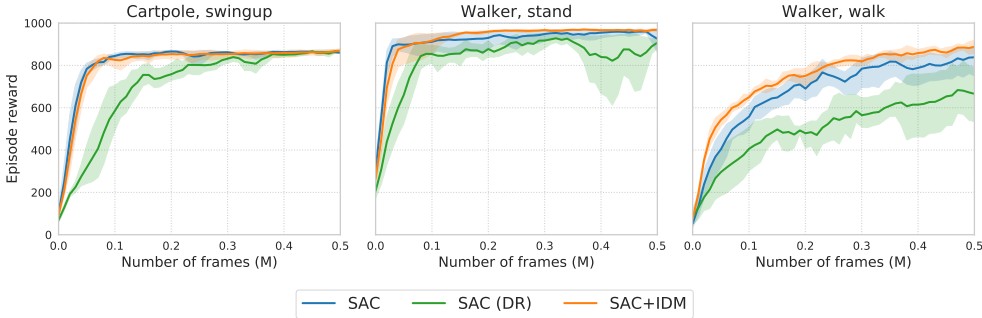

*Figure 5.* Learning curves for SAC, SAC trained with domain randomization (denoted *SAC (DR)* here), and SAC+IDM on three tasks from the DeepMind Control suite (DMControl). Episodic return is averaged across 10 seeds and the 95% confidence intervals are visualized as shaded regions. SAC and SAC+IDM exhibit similar sample efficiency and final performance, whereas domain randomization consistently displays worse sample efficiency, larger variation between seeds, and converges to sub-optimal performance in two out of the three tasks shown.

## C  KEEPING $\pi_s$ FIXED DURING POLICY ADAPTATION

We now consider a variant of PAD where the self-supervised task head $\pi_s$ is fixed at test-time such that the self-supervised objective $L$ is optimized only wrt $\pi_e$, as discussed in Section 3.3. We measure generalization to test environments with randomized colors and report the results in Table 9 for three tasks from the DeepMind Control suite. We empirically find the difference between updating $\pi_s$ and keeping it fixed negligible, and we choose to update $\pi_s$ by default since its gradients are computed by back-propagation regardless.

*Table 9.* Episodic return in test environments with randomized colors, mean and std. dev. for 10 seeds. All methods use SAC. *IDM (PAD, fixed $\pi_s$)* considers a variant of PAD where $\pi_s$ is fixed at test-time, whereas *IDM (PAD)* denotes the default usage of PAD in which both $\pi_e$ and $\pi_s$ are optimized at test-time using the self-supervised objective.

| Random colors | IDM | IDM (PAD, fixed $\pi_s$) | IDM (PAD) |
|---|---|---|---|
| Walker, walk | $406{\pm}29$ | $452{\pm}38$ | $468{\pm}47$ |
| Walker, stand | $743{\pm}37$ | $802{\pm}41$ | $797{\pm}46$ |
| Cartpole, swingup | $585{\pm}73$ | $623{\pm}57$ | $630{\pm}63$ |

## D  COMPARISON TO ADAPTATION WITH REWARDS

While our method does *not* require data collected prior to deployment and does *not* assume access to a reward signal, we additionally compare our method to a naïve fine-tuning approach using transitions and rewards collected from the target environment prior to deployment. To fine-tune the pre-trained policy using rewards, we collect datasets consisting of 1, 10, and 100 episodes in each target environment using the learned policy while keeping its parameters fixed, and then subsequently fine-tune both $\pi_e$ and $\pi_a$ on the collected data, following the same training procedure as during the training phase. This fine-tuning approach is analogous to Julian et al. (2020) but does not use data from the original environment during adaptation. Results are shown in Table 10. We find that naïvely fine-tuning the policy using data collected prior to deployment can improve generalization but requires comparably more data than PAD, as well as access to a reward signal in the target environment. This finding suggests that PAD may be a more suitable method for settings where data from the target environment is scarce and not easily accessible prior to deployment.

## E  ADDITIONAL ROBOTIC MANIPULATION SAMPLES

Figure 6 provides samples from the training and test environments for the *reach* robotic manipulation task. Agents are trained in simulation and deployed on a real robot. Samples from the *push* task are shown in Figure 4.

*Table 10.* Episodic return in test environments with randomized colors, mean and std. dev. for 10 seeds. All methods use SAC trained with an inverse dynamics model (IDM) as auxiliary task. Our method is denoted *IDM (PAD)*, and we compare to a naïve fine-tuning approach that assumes access to transitions and rewards collected from 1, 10, and 100 episodes, respectively, from target environments *prior* to deployment.

| | | | Fine-tuning w/ rewards | | |
|---|---|---|---|---|---|
| Random colors | IDM | IDM (PAD) | 1 episode | 10 episodes | 100 episodes |
| Walker, walk | $406\pm29$ | $468\pm47$ | $395\pm78$ | $489\pm104$ | $561\pm62$ |
| Walker, stand | $743\pm37$ | $797\pm46$ | $661\pm65$ | $728\pm44$ | $784\pm31$ |
| Cartpole, swingup | $585\pm73$ | $630\pm63$ | $538\pm53$ | $605\pm51$ | $650\pm58$ |

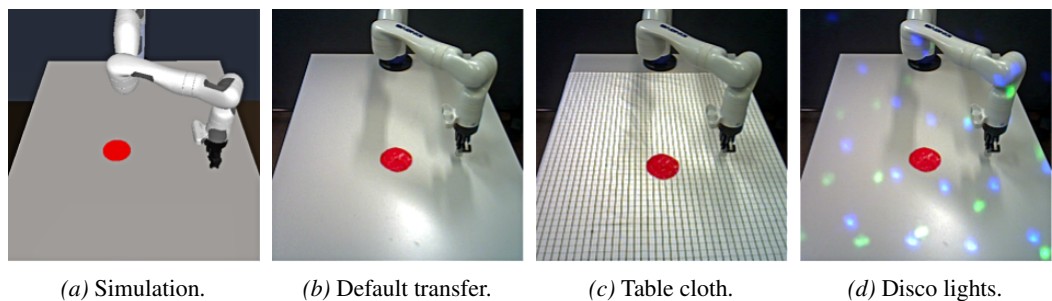

| *(a)* Simulation. | *(b)* Default transfer. | *(c)* Table cloth. | *(d)* Disco lights. |
|---|---|---|---|

*Figure 6.* Samples from the *reach* robotic manipulation task. The task is to move the robot gripper to the location of the red disc. Agents are trained in setting (a) and evaluated in settings (b-d) on a real robot, taking observations from an uncalibrated camera.

## F    IMPLEMENTATION DETAILS

In this section, we elaborate on implementation details for our experiments on DeepMind Control (DMControl) suite (Tassa et al., 2018) and CRLMaze (Lomonaco et al., 2019) for ViZDoom (Wydmuch et al., 2018). Our implementation for the robotic manipulation experiments closely follows that of DMControl. Code is available at https://nicklashansen.github.io/PAD/.

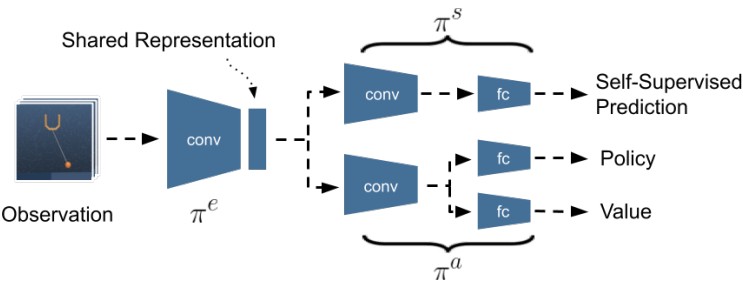

*Figure 7.* Network architecture for the DMControl, CRLMaze, and robotic manipulation experiments. $\pi^s$ and $\pi^a$ uses a shared feature extractor $\pi^e$. Observations are stacks of $100 \times 100$ colored frames. Implementation of policy and value function depends on the learning algorithm.

**Architecture.**    Our network architecture is illustrated in Figure 7. Observations are stacked frames ($k = 3$) rendered at $100 \times 100$ and cropped to $84 \times 84$, i.e. inputs to the network are of dimensions $9 \times 84 \times 84$, where the first dimension indicates the channel numbers and the following ones represent spatial dimensions. The same crop is applied to all frames in a stack. The shared feature extractor $\pi^e$ consists of 8 (DMControl, robotic manipulation) or 6 (CRLMaze) convolutional layers and outputs features of size $32 \times 21 \times 21$ in DMControl and robotic manipulation, and size $32 \times 25 \times 25$ in CRLMaze. The output from $\pi^e$ is used as input to both the self-supervised head $\pi^s$ and RL head $\pi^a$, both of which consist of 3 convolutional layers followed by 3 fully-connected layers. All

*Table 11.* Hyperparameters used for the DM-Control (Tassa et al., 2018) tasks.

| Hyperparameter | Value |
| --- | --- |
| Frame rendering | $3 \times 100 \times 100$ |
| Frame after crop | $3 \times 84 \times 84$ |
| Stacked frames | 3 |
| Action repeat | 2 (finger) |
| | 8 (cartpole) |
| | 4 (otherwise) |
| Discount factor $\gamma$ | 0.99 |
| Episode length | 1,000 |
| Learning algorithm | Soft Actor-Critic |
| Self-supervised task | Inverse Dynamics Model |
| Number of training steps | 500,000 |
| Replay buffer size | 500,000 |
| Optimizer ($\pi^e, \pi^a, \pi^s$) | Adam ($\beta_1 = 0.9, \beta_2 = 0.999$) |
| Optimizer ($\alpha$) | Adam ($\beta_1 = 0.5, \beta_2 = 0.999$) |
| Learning rate ($\pi^e, \pi^a, \pi^s$) | 3e-4 (cheetah) |
| | 1e-3 (otherwise) |
| Learning rate ($\alpha$) | 1e-4 |
| Batch size | 128 |
| Batch size (test-time) | 32 |
| $\pi^e, \pi^s$ update freq. | 2 |
| $\pi^e, \pi^s$ update freq. (test-time) | 1 |

*Table 12.* Hyperparameters used for the CRL-Maze (Lomonaco et al., 2019) navigation task.

| Hyperparameter | Value |
| --- | --- |
| Frame rendering | $3 \times 100 \times 100$ |
| Frame after crop | $3 \times 84 \times 84$ |
| Stacked frames | 3 |
| Action repeat | 4 |
| Discount factor $\gamma$ | 0.99 |
| Episode length | 1,000 |
| Learning algorithm | Advantage Actor-Critic |
| Self-supervised task | Rotation Prediction |
| Number of training episodes | 1,000 (dom. rand.) |
| | 500 (otherwise) |
| Number of processes | 20 |
| Optimizer | Adam ($\beta_1 = 0.9, \beta_2 = 0.999$) |
| Learning rate | 1e-4 |
| Learning rate (test-time) | 1e-5 |
| Batch size | 20 |
| Batch size (test-time) | 32 |
| $\pi^e, \pi^s$ loss coefficient | 0.5 |
| $\pi^e, \pi^s$ loss coefficient (test-time) | 1 |
| $\pi^e, \pi^s$ update freq. | 1 |
| $\pi^e, \pi^s$ update freq. (test-time) | 1 |

convolutional layers use 32 filters and all fully connected layers use a hidden size of 1024, as in Yarats et al. (2019).

**Learning algorithm.** We use Soft Actor-Critic (SAC) (Haarnoja et al., 2018) for DMControl and robotic manipulation, and Advantage Actor-Critic (A2C) for CRLMaze. Network outputs depend on the task and learning algorithm. As the action spaces of both DMControl and robotic manipulation are continuous, the policy learned by SAC outputs the mean and variance of a Gaussian distribution over actions. CRLMaze has a discrete action space and the policy learned by A2C thus learns a soft-max distribution over actions. For details on the critics learned by SAC and A2C, the reader is referred to Haarnoja et al. (2018) and Mnih et al. (2016), respectively.

**Hyperparameters.** When applicable, we adopt our hyperparameters from Yarats et al. (2019) (DM-Control, robotic manipulation) and Lomonaco et al. (2019) (CRLMaze). For the robotic manipulation experiments, our implementation closely follows that of DMControl, only differing by number of frames in an observation. We use a frame stack of $k = 3$ frames for DMControl and CRLMaze, and only $k = 1$ frame for robotic manipulation. For completeness, we detail all hyperparameters used for the DMControl and CRLMaze environments in Table 11 and Table 12.

**Data augmentation.** Random cropping is a commonly used data augmentation used in computer vision systems (Krizhevsky et al., 2012; Szegedy et al., 2015) but has only recently gained interest as a stochastic regularization technique in the RL literature (Srinivas et al., 2020; Kostrikov et al., 2020; Laskin et al., 2020). We adopt the random crop proposed in Srinivas et al. (2020): crop rendered observations of size $100 \times 100$ to $84 \times 84$, applying the same crop to all frames in a stacked observation. This has the added benefits of regularization while still preserving spatio-temporal patterns between frames. When learning an inverse dynamics model, we apply the same crop to all frames of a given observation but apply two different crops to the consecutive observations $(\mathbf{s}_t, \mathbf{s}_{t+1})$ used to predict action $\mathbf{a}_t$.

**Policy Adaptation during Deployment.** We evaluate our method and baselines by episodic return of an agent trained in a single environment and tested in a collection of test environments, each with distinct changes from the training environment. We assume no reward signal at test-time and agents are expected to generalize without pre-training or resetting in the new environment. Therefore, we make updates to the policy using a self-supervised objective, and we train using observations from the environment in an online manner without memory, i.e. we make one update per step using the most-recent observation.

Empirically, we find that: (i) the random crop data augmentation used during training helps regularize learning at test-time; and (ii) our algorithm benefits from learning from a batch of randomly cropped observations rather than single observations, even when all observations in the batch are augmented copies of the most-recent observation. As such, we apply both of these techniques when performing Policy Adaptation during Deployment and use a batch size of 32. When using the policy to take actions, however, inputs to the policy are simply center-cropped.

