# OpenReview forum: "Self-Supervised Policy Adaptation during Deployment"
_ICLR.cc/2021/Conference — ICLR 2021 Spotlight_

### Official Review · AnonReviewer4 · 2020-10-28
**Official Blind Review**

**Rating:** 7
**Confidence:** 4

**Review:**

This paper studies an important problem in vision-based RL: how to adapt a pre-trained policy to an unseen environment in a self-supervised manner. To do this, the authors introduce an auxiliary task branch that can be used to tune the intermediate representation of the policy network on the fly in a self-supervised manner(e.g. inverse dynamic prediction). The experiments in the DeepMind Control suite, CRLMaze, and robot manipulation tasks show the generalization and effectiveness of the proposed method in various vision-based RL problems.

Strength:
- The paper is well-written and easy-to-follow. The demo video is impressive.
- The problem they studied is important to the RL community, especially for virtual2real adaption.
- The proposed method is simple yet effective. The author conducts experiments in various environments and provides a comprehensive analysis of the results. The implementation details are also detailed for reproduction.

Weakness:
- Lack of novelty. The proposed method can be viewed as a simple application of the self-supervised auxiliary task in the domain adaptation.  Every component used in the proposed method is already introduced in previous work. It is trivial to select the auxiliary task for different environments.
-  The setting. It seems that the training environment and the test environment only differ in appearance. However, in real-world robot settings, the test environment would be of unseen transition dynamic, e.g. additional noise in action, delay in the control loop. How does this method perform in this situation?

---

> ### Author Response · Authors · 2020-11-19
> **Response to reviewer #4 - general thanks and comments**
>
> Dear reviewer #4, thank you for your comments. We have updated our submission with a number of minor changes that seek to provide more clarity. We have also made a general comment to provide further discussion and insights into our method, as well as a list of our changes. In the following, we seek to address each of your concerns.
>
> **Q:** *“Lack of novelty. The proposed method can be viewed as a simple application of the self-supervised auxiliary task in the domain adaptation. Every component used in the proposed method is already introduced in previous work.”*
>
> **A:** We respectfully disagree with the statement that our proposed method is not novel, and actually consider the simplicity of our method one of its key strengths. This aspect of simplicity has also been highlighted by the other reviewers: *“The authors present a novel, intuitive method for domain transfer in the case of absence of reward signal.”* (R1); *“This work's originality stems primarily from its simplicity. I commend the authors for avoiding overly-complicated methods when simpler ones will do.”* (R2).
> We argue that our paper offers a novel perspective on how to adopt RL policies on unseen and changing environments by adapting the policy during test time on the fly, instead of using domain randomization. Our adaptive method can be applied to various environments including Sim2Real robot manipulation *in a continuously changing environment*, which itself is a problem setting that has barely been addressed in previous work. We believe this is a realistic and important problem to address in RL and robotics.
>
> While self-supervised learning has shown promising results in different areas, it has never been applied in the problem setting of adapting RL policies to various (and continuously changing) environments in an online manner. We stress the fact that our simple method can be generalized to applications including motor control, navigation, robotic manipulation is a non-trivial discovery.
>
> **Q:** *" in real-world robot settings, the test environment would be of unseen transition dynamic, e.g. additional noise in action, delay in the control loop. How does this method perform in this situation?"*
>
> **A:** Deployment on real-world robots is an important setting to consider, as it is in fact one of the key settings in which rewards may not be available at test-time. In section 4.3, we consider the Sim2Real setting by training an agent for robotic manipulation in simulation and transfer it to a real robot. The real-world environment differs from the simulation, both visually and in terms of dynamics, such as object dimensions, mass, friction, and uncalibrated actions. While we primarily consider perceptual adaptation in this work, we find our method to consistently improve in each of the real-world test environments, including the push task that features considerable object interaction (Table 5). We elaborate on this in our general response and provide an additional experiment in which we also consider adaptation to changes only in dynamics (Table 6). We have updated the paper to better emphasize these points.

---

> > ### Comment · AnonReviewer4 · 2020-11-22
> > **Reply to authors' response**
> >
> > Thank you for your detailed responses to my comments and questions. And I have revised my score to 7, based on the convincing results in Table. 6, which demonstrates the potential of PAD in unseen dynamics. I recognize that it is the first work that applies simple/standard auxiliary tasks (inverse dynamic model) for policy adaption without a reward function.  But I still stand that the novelty of the proposed method is marginal, as it is only a standard application of an existing method in an unexplored setting.
> > Even though the method is not novel,  the problem they studied is important and the empirical results are interesting and valuable to the community. So, I agree to accept this paper.

---

> ### Comment · AnonReviewer2 · 2020-11-21
> **Combining components and showing they work well is novelty**
>
> I fundamentally disagree that the work is not novel because it is not surprising, and uses components introduced in previous works.
>
> What was the point of previous authors introducing those components, except for others to build off of them? Disincentivizing authors from building on the work of others, because they fear retribution from reviewers for not reinventing the wheel, is bad for the community.
>
> This paper shows that a simple auxiliary task--yes, introduced in previous works--is effective as a self-supervised adaptation method, and that method works with real robots, not just simulated (i.e. pretend) robots. Showing that this works well for RL, and transfers to real robots, is more than enough of a contribution.
>
> Surprise is not the goal of science, understanding is. Data is not the plural of assumptions, and showing that something works in an important application domain, measuring exactly how well it works against alternatives, and showing others how to do it in the future is absolutely novel. This gives future researchers confidence that they can build on this work.

---

### Official Review · AnonReviewer1 · 2020-10-28
**Overall strong submission but missing some discussion**

**Rating:** 7
**Confidence:** 4

**Review:**

#### Summary
The authors present a method for online policy adaptation during domain transfer in the case no reward is available in the target domain. They achieve this by adding an auxiliary self-supervised task, such as inverse dynamics prediction, that helps shape a set of features shared with the policy during training. At test time, gradient updates are then performed on these shared features based on only the self-supervised loss. The authors evaluate their method on a number of visual continuous control tasks and discrete navigation tasks and show a significant improvement over direct transfer, domain randomisation, and using self-supervision only during training.

#### Pros
- The authors present a novel, intuitive method for domain transfer in the case of absence of reward signal. This is very relevant for e.g. sim2real transfer, where reward computation often require privileged information that is not easily measurable in a real-world setup.
- The authors perform a large sweep of experiments on a variety of different tasks, incl. sim2real transfer on a robot arm, and provide significant detail of the experimental setup.
- The results are largely in favour of the proposed method in comparison to fair baselines.
- The paper is very well and clearly written, with significant attention to related work.

#### Cons
- My main concern with this submission is the lack of any discussion about the assumptions and resulting limitations of the proposed method. It is only in Section 3 that it becomes clear that this method is aimed at _perceptual_ adaptation, but otherwise assumes the transition dynamics of the source and target environment, and hence the resulting optimal action, to be the same. It's unclear how this method fairs in the case of a change in dynamics. This should definitely be more emphasised in the introduction, abstract and perhaps even title. This also puts the proposed method closer to methods such as RCAN (https://arxiv.org/abs/1812.07252), which would potentially be a strong baseline to compare to.
- While the authors acknowledge that drift in feature space may occur by also updating the self-supervision-only weights at test time, they default to this setting but do not show any quantitative results and only mention that empirically this indeed does not significantly impact performance. Perhaps related, but the IDM+PAD setting seems to be is significantly detrimental in the CRLMaze task (see Table 4, Table 7), it would good to add a comment about this failure mode.

#### Conclusion
While there could be some additional discussion added, I think this paper is strong enough to be accepted.

---

> ### Author Response · Authors · 2020-11-19
> **Response to reviewer #1 - general thanks and comments**
>
> Dear reviewer #1, we thank you for your suggestions and insights. In light of your comments (and those of your fellow reviewers) we have revised our paper and highlighted changes in red. We have also made a general comment to provide further discussion and insights into our method, as well as a list of our changes. In the following, we seek to address each of your concerns.
>
> **Q:** *“This also puts the proposed method closer to methods such as RCAN (https://arxiv.org/abs/1812.07252), which would potentially be a strong baseline to compare to.”*
>
> **A:** First of all, thank you for your suggestion. Unfortunately, we found RCAN difficult to reproduce since their implementation was not open-sourced. While it would be an interesting comparison, we believe that the method is in a similar spirit to our domain randomization baseline, and ultimately suffers from the same problem of distributional shift that we seek to address (see also our general response for further discussion on this issue). We have added RCAN to related work in our revised version.
>
> **Q:** *“[...] drift in feature space may occur by also updating the self-supervision-only weights at test time, they default to this setting but do not show any quantitative results [...]”*
>
> **A:** In single-step optimization, the two (fixed or not) are equivalent. In multi-step optimization like PAD, we empirically find little to no difference. We conjecture that such “drifting” of features does not occur because 1) two tasks are jointly optimized during training; and 2) their gradients are sufficiently correlated. Below, we provide comparisons on three tasks from DeepMind Control in the randomized color test environment (mean and standard deviation of 10 seeds).
>
> | Random colors     |   IDM  |   IDM (PAD, fixed pi_s)   | IDM (PAD) |
> |-------------------|:------:|:-------------------------:|:---------:|
> | Walker, walk      | 406±29 |           452±38          |   468±47  |
> | Walker, stand     | 743±37 |           802±41          |   797±46  |
> | Cartpole, swingup | 585±73 |           623±57          |   630±63  |
>
> We have added these results to appendix (section C) in the revised version and referenced them in section 3. In general we conjecture that, while an auxiliary task can enforce structure in the learned representations, its features (and consequently gradients) need to be sufficiently correlated with the primary task for PAD to be successful. If it is not, the auxiliary task may negatively impact representation during both training and testing, e.g. on DeepMind Control the rotation prediction baseline generalizes poorly compared to the inverse dynamics model, and likewise adaptation with rotation prediction does not improve significantly either (see Table 3). For the CRLMaze environment, we observe the same phenomenon but the other way around (see Table 4). We discuss this further in section 4.1 and 4.2.

---

### Official Review · AnonReviewer3 · 2020-10-28
**Pixel variations in sim2real can be compensated for with auxiliary tasks.**

**Rating:** 7
**Confidence:** 4

**Review:**

This paper presents the Policy Adaptation during Deployment (PAD) method, which allows a policy trained on a particular visual input distribution to be transferred to a new visual input distribution, as long as the underlying system dynamics remain the same.

PAD works by structuring the policy into two parts, a state embedding network pi_e and an action selection network pi_a.  Additionally, PAD learns an auxiliary prediction task with a network pi_s that predicts the inverse dynamics (necessary action to transition from s->s').  Once on the real system, pi_a is kept frozen, and pi_e is updated to minimize the inverse dynamics loss.  In practice it seems that also updating pi_s provides negligible difference although I find only the case when pi_s is kept frozen to make sense (question on this below).  An alternative loss based on predicting scene rotation is also used and shown to work generally less well than inverse dynamics.

Experiments show that PAD consistently outperforms other methods, either naive baselines, simplified domain randomisations, or ablations of PAD.

Pros: I enjoyed reading the paper and believe it provides an interesting approach to used IDM as a consistent aspect of the task in the face of inconsistent visual representations of state.

Cons: This approach is very specific to pixels and assumes consistent inverse dynamics, which might not always be true esp. in sim2real tasks.    I think the more general question is how to adapt in MDPs when part of the MDP changes but another part stays constant.  The 'trick' in my opinion in this paper is that the dynamics are consistent across task variations, so building a predictor for this consistent dimension works.  It would have been interesting to look more generally at other aspects that could change, for example what if dynamics change but pixel representations are consistent, could a similar approach be used?  Is there a 'bigger' insight here?  I am overall accepting the paper, but I feel a bit frustrated we're not getting more insights and I feel like there's something more here.

Questions:
I am clearly being dense here, but I wasn't able to understand why the constant zero feature would be a trivial solution to a next-state prediction problem (c.f.
" Note that we predict the inverse dynamics instead of the forward dynamics, because
when operating on the feature space, the latter can produce trivial solutions such as the constant zero
feature for every state."), could you clarify?

Although you mention that updating both pi_s and pi_e or only pi_e seems to provide equivalent performance, intuitively it would seem that allowing pi_s to move removes the common semantic grounding of the latent observation space that is necessary for pi_a to still be able to interpret it.  Do you have any insights here?

Please comment on my 'con', and provide some insights on how you see this approach working in potentially other dimensions than the observation space.

---

> ### Author Response · Authors · 2020-11-19
> **Response to reviewer #3 - general thanks and comments**
>
> Dear reviewer #3, thank you for your valuable comments. In light of your comments (and those of your fellow reviewers) we have revised our paper and highlighted changes in red. We have also made a general response to provide further discussion and insights into our method, as well as a list of our changes. In the following, we seek to address each of your concerns.
>
> **Q:** *“This approach is very specific to pixels and assumes consistent inverse dynamics, which might not always be true esp. in sim2real tasks.”*
>
> **A:** We would like to emphasize that we do not assume consistent dynamics, we simply assume that the task objective remains the same. As you point out, the dynamics in sim2real settings are seldom consistent even though the task itself remains unchanged. We elaborate on this in the general response, and have made minor updates to the paper to clarify this.
>
> **Q:** *“ why the constant zero feature would be a trivial solution to a next-state prediction problem”*
>
> **A:** Given a state and an action, a Forward Dynamics Model (FDM) predicts the next state. When operating in pixel space, an FDM predicts the full image observation regardless of content, which is often found to be a poor modeling choice since the majority of the image may be irrelevant to the task. An alternative is to operate in the (latent) feature space, such that the FDM predicts the latent representation of the next state rather than the image observation itself. This formulation of the FDM, however, has the trivial solution that learning to map any (observation, action) pair into a constant feature vector (e.g. the 0-vector) yields a prediction loss of 0. Because an inverse dynamics model does not have such trivial solutions, we opt for that instead. We have added a footnote in the revised version of the paper that clarifies this.
>
> **Q:** *“Although you mention that updating both pi_s and pi_e or only pi_e seems to provide equivalent performance, intuitively it would seem that allowing pi_s to move removes the common semantic grounding of the latent observation space that is necessary for pi_a to still be able to interpret it. Do you have any insights here?”*
>
> **A:** In single-step optimization, the two (fixed or not) are equivalent. In multi-step optimization like PAD, we empirically find little to no difference. We conjecture that such “drifting” of features does not occur because 1) two tasks are jointly optimized during training; and 2) their gradients are sufficiently correlated. We have added an additional experiment to appendix (section C) that empirically shows the two formulations yield similar results, and referenced it in Section 3.

---

### Official Review · AnonReviewer2 · 2020-11-02
**A high quality contribution with a couple easy-to-address blind spots**

**Rating:** 7
**Confidence:** 4

**Review:**

## Summary of the Work
This work presents a method for test-time adaptation of pre-trained visual reinforcement learning policies, with a particular emphasis on robotics applications. The method uses both a self-supervised representation learning objective and an RL objective during training, then at test-time (when RL supervision may not be available) adapts the agent to a new environment using only the self-supervision objective. The self-supervision objective is to learn an inverse dynamics model $p(s_t|a_{t-1}, s_{t-1})$ using an IDM network which shares its convolutional layers with the policy network $p(a_t|s_t)$. The authors provide experiments using a variety of simulation environments augmented with adaptation target tasks, then demonstrate the effectiveness of the method using real robot experiments. Overall, the method appears to be a simple and effective idea for quick test-time adaptation to appearance changes in visual RL, though the lack of a fine-tuning baseline makes it unclear how it compares to the "null" hypothesis of doing naive RL updates with no auxiliary losses

## Pros and Cons

### Pros
* Addresses an important problem in the field (fast test-time adaptation)
* Extensive experiments supporting claims, including real robot experiments

### Cons
* Likely limited only to appearance changes in the environment (not dynamics nor reward functions)
* Comparisons to more "naive" fine-tuning baseline missing -- this compares favorably to other self-supervision systems, but how does it compare to the most basic adaptation method available to an RL agent?

## Evaluation
### Quality
3/5
The overall presentation quality of the work is very high, but I take issue with some of the scientific aspects. While the experiments are all high-quality and well thought-out, this work has some issues with data analysis which are easy to address, and would make the statistical significance of the results much more clear. Why go to the trouble of collecting 10 seeds for many experiments, then only calculate a mean and standard deviation? I also believe a couple obvious baselines (fine-tuning and from-scratch training) were unnecessarily omitted and would help readers better-position the performance of this work as compared to the "naive" alternatives.

### Clarity
5/5
Every part of the text and supplemental were very well-presented. The tables and figures are helpful for understanding the method and results, and I found the mathematical portions of the text increase the clarity of presentation rather than simply providing a "window dressing" of formalism intended to make the work seem more principled than it really is.

### Originality
4/5
This work's originality stems primarily from its simplicity. I commend the authors for avoiding overly-complicated methods when simpler ones will do.

### Significance
4/5
Addressing test-time adaptation without costly pre-train methods (e.g. meta-learning) is an important challenge for RL in the real world, and this work provides a step in this direction.  I would also appreciate if the work spent more time discussing the limitations of the method, and avoided statements which border on over-stating its potential -- for instance, the introduction suggests that adapting to new reward functions at test time is important (it is!) but this text contains no evidence that the proposed method could achieve that kind of adaptation.


### Misc Editorial Comments and Reviewer's Notes

#### Claims
* Adapts a pre-trained policy to an unknown environment without any reward

#### Mechanisms
* Introduces self-supervision supervised with an auxiliary task
* Auxiliary tasks
  - inverse dynamics prediction
  - rotation prediction (classify rotation of image)
* Auxiliary prediction networks share parameters with the policy (particularly, CNN layers)
* Trains both RL and auxiliary task during training, at test time uses IDM or rotation prediction auxiliary updates to adapt (but not RL)


#### 1. Introduction
* The reward functions in these experiments don't seem to change, so I don't think the motivation about "craft[ing] a dense reward function...during deployment is impractical" really applies? This motivation would be more compelling if the work proposed adaptation to new reward functions rather than only visual changes to the environment. Adaptation to new reward functions and dynamics likely requires a different family of methods than those addressed by this work. Again, see [1].


#### 2. Related Work
* Please see [1] for a similar work with a similar experimental setting. In particular, this calls into question the conventional wisdom that fine-tuning is too naive for test-time adaptation and can be immediately discarded with a citation to Rusu, et al. Please consider including a baseline (at least in simulation) comparing the performance of PAD to "naive" fine-tuning or NSL-style fine-tuning (oversampling).

#### 4. Experiments
* " as we find that learning a model of the motors works well for motor control." I didn't understand this comment. Please make this more clear -- are you introducing a modeling prior which attempts to explicitly model the actuators and joints of these models?
* Table 2, 3, 4: I find it misleading to highlight elements of the table which don't exceed the performance of the other elements *including their confidence intervals*. Statistically, these entries might be equivalent, with some probability. However, the simple standard deviation from the mean is a poor confidence interval for comparisons-- please instead include a CI calculated with a bootstrap method and a known p-value (e.g. 95% interval) to enable better comparison. seaborn can do this for you. See [2] for a guide.
* Table 5: Please include a confidence interval
* Would prefer to see an ablation of the test-time performance which includes only the RL loss and RL loss+IDM(PAD), in addition to the existing "no RL loss + IDM(PAD)". Additionally, it would be helpful to show the reader the performance of a scratch-trained policy on the test environments, to establish an "oracle" or maximum expected performance for each environment given your base algorithms (A2C and SAC).

#### Appendix
* Figure 5: Please calculate a confidence interval for these curves with a known p-value (e.g. 95%) rather than simple standard deviation. This makes it much easier for readers to assess the statistical significance of your results.


[1] https://arxiv.org/abs/2004.10190
[2] https://arxiv.org/abs/1904.06979

---

> ### Author Response · Authors · 2020-11-19
> **Response to reviewer #2 - general thanks and comments**
>
> Dear reviewer #2, thank you for your detailed and thorough review. In light of your comments (and those of your fellow reviewers) we have revised our paper and highlighted changes in red. We have also made a general response to provide further discussion and insights into our method, as well as a list of our changes. In the following, we seek to address each of your concerns.
>
> **Q:** *“Comparisons to more "naive" fine-tuning baseline”*
>
> **A:** In this work, we consider the problem setting in which rewards are unavailable during deployment, which is generally true for real-world deployment of agents trained by reinforcement learning. When rewards are not available, we generally have two options: 1) construct a new reward signal; or 2) adapt without rewards. A way to obtain a sparse reward signal in the real world could be by manually observing the robot’s behavior and giving a positive reward when it succeeds. Alternatively, a separate system can be built to automatically detect some predefined success criteria, similar to NSL and QT-Opt [1]. This separate system, however, is equally prone to distributional shift during deployment, which can corrupt the reward signal. Even assuming a consistent reward signal, fine-tuning by rewards **requires manual labor, consequently lacking generality**, may require a significant number of trials (e.g. NSL uses 800 trials in the test environment whereas our method requires none), and cannot effectively adapt to non-stationary environments. We therefore argue that adaptation without rewards is preferred in situations where the ground-truth reward signal is unavailable. We have updated the introduction to better reflect this.
>
> **Q:** *“I also believe a couple obvious baselines (fine-tuning and from-scratch training) were unnecessarily omitted and would help readers better-position the performance of this work as compared to the "naive" alternatives.”*
>
> **A:** We agree that a fine-tuning baseline can help better position the performance of our method; we will add it to the paper. Additionally, we would like to emphasize that in the DeepMind Control and CRLMaze experiments, we report both the fixed environment training performance (Table 7 and 8) and domain randomization baselines, which also serve to help gauge the performance of our method. To make the domain randomization baseline a more fair comparison, colors are sampled from the same color distribution as in the randomized color test environment, but with lower variance, as we find that training directly on the test distribution is unstable and converges to sub-par policies.
>
> **Q:** *“Why go to the trouble of collecting 10 seeds for many experiments, then only calculate a mean and standard deviation?”*
>
> **A:** We follow the evaluation protocol of prior work for both DeepMind Control (Yarats et al., 2019; Srinivas et al., 2020; Laskin et al., 2020) and CRLMaze (Lomonaco et al., 2019), which all report mean and standard deviation across 10 seeds. Reporting standard deviation can help gauge the variability of methods such as domain randomization that are prone to unstable training, and following the same protocol as prior work makes comparison easier. We have updated Figure 5 to show the 95% confidence interval of methods rather than the standard deviation.
>
> **Q:** *" as we find that learning a model of the motors works well for motor control."*
>
> **A:** We are learning a model of the dynamics -- this has now been clarified in the updated paper draft.
>
> [1] https://arxiv.org/abs/1806.10293

---

> > ### Comment · AnonReviewer2 · 2020-11-21
> > **Thank you and revised score**
> >
> > Thank you for the thoughtful and thorough response to my comments and questions.
> >
> > I appreciate your perspective on how this compares to fine-tuning alternatives, and would encourage you to discuss this trade-off further for the reader in a final version. A quick note: there's no reason to believe fine-tuning can't deal with non-stationarity if it is applied repeatedly.
> >
> > Considering your revisions, I've updated my revised score to "7: Good paper, accept" and I think the paper would be even more compelling once you add the promised fine-tuning baselines to the draft.

---

### Author Response · Authors · 2020-11-19
**General response and summary of revision**

Dear reviewers, we appreciate all your feedback and have made a number of minor changes to address your comments. We have highlighted changes in red in the revised version, and also provide an overview of our changes in the following.

**Discussion on assumptions**
While it is true that we primarily consider adaptation to perceptual differences in the environment, we want to emphasize that our method does not assume the transition dynamics of the source and target environments remain the same, only that the task’s objective itself is unchanged. Since one of the self-supervised tasks that we consider is in fact an inverse dynamics model, our method can be used for adaptation to low-level changes in the dynamics, although this work mainly considers visual adaptation.
For instance, in the Sim2Real setting that we consider, the simulation (training environment) and real robot (test environments) differ not only visually, but also in terms of dynamics. Notable differences include physical properties such as object dimensions, mass, friction, as well as subtle differences in end effector movement since actions are uncalibrated. On the push task, we observe substantial improvements with our method compared to baselines, which we conjecture may be due to the additional difficulty introduced by object interaction under minor changes in dynamics.

We have added an additional robotic manipulation experiment to the revised version of the paper, where we consider adaptation to changes in dynamics only (Table 6, also shown below). *Push (object)* considers change of object dimensions, mass, and friction, *Push (mount)* considers the robotic arm mounted in a different position, *Push (velocity)* considers change in end effector velocity, and *Push (all)* considers all 3 modifications jointly.

| Simulated robot | SAC | +DR | +IDM | +IDM (PAD) |
|-----------------|:---:|:---:|:----:|:----------:|
| Push (object)   | 66% | 64% |  72% |     **82%**    |
| Push (mount)    | 68% | 58% |  **86%** |     84%    |
| Push (velocity) | 70% | 68% |  70% |     **78%**    |
| Push (all)      | 56% | 50% |  48% |     **76%**    |

We find that our method can adapt to subtle changes in dynamics, similar to those present in our Sim2Real experiments. Self-supervised adaptation to drastically different (and potentially non-stationary) dynamics may therefore be an interesting avenue for future research, and could lead to a new suite of methods that also adapt pi_a at test-time. We have made these points more clear in the revised version of the paper.


**List of changes:**
- Added a footnote clarifying why a forward dynamics model operating in feature space has trivial solutions (Page 4).
- Added experiment to Section 4.3 on adaptation to dynamics.
- Added experiment to appendix (section C) where pi_s is fixed at test-time.
- More clearly state differences between the simulation and real robot deployment.
- Updated introduction to better reflect related work on fine-tuning with rewards.
- Added RCAN and NSL to related work.
- “learning a model of the motors” -> “learning a model of the dynamics”
- Update Figure 5 (learning curves) in appendix to show 95% confidence intervals as shaded regions instead of standard deviation.
- Subtle changes in wording and presentation to make details more clear

---

### Decision · Program_Chairs · 2021-01-07
**Final Decision**

**Decision:**

Accept (Spotlight)

**Comment:**

This paper describes a method for adapting an RL policy in a deployment environment that does not provide a reward signal.  This concern arises commonly when a task reward is available in a robot simulator but not on the physical robot where the policy is eventually deployed.  The proposed solution is to learn an inverse dynamics model as an auxiliary prediction task on an internal state embedding that is shared with the policy.  The policy is adapted during deployment by modifying the state embedding using this auxiliary task (with the assumption that the main objective remains unchanged).  The proposed method is tested with transfer between simulated domains and also on transfer from a simulator to a physical robot.  The experiments showed the method had consistently higher performance than alternatives.

The reviewers found many positive contributions in the presented paper. These include the problem's importance (R1, R2,R4), extensive experiments (R1, R2, R3), clear writing (R1,R4), simplicity and effectiveness in comparison to ablations (R3, R4).  The reviewers saw a weakness in the method's limitation to perceptual adaptation instead of dynamics adaptation (R1-4) and the lack of novelty (R4).  The author response addressed both concerns.  They stated that the method is novel for adapting to continuously changing environments in a self-supervised fashion without rewards.  The authors modified the paper to clarify how the method demonstrates robustness to changes in the system dynamics.  The reviewers found the author response addressed their major concerns.

Four reviewers indicate accept for the contributions stated above and expressed no remaining concerns.  The paper is therefore accepted.